# Distribution of Genetic Diversity in *Beta patula* Aiton Populations from Madeira Archipelago, Portugal

Carla Ragonezi [1,2,*], Humberto Nóbrega [1], Maria Inês Leite [1], José G. R. de Freitas [1], Fabrício Lopes Macedo [1,2] and Miguel Â. A. Pinheiro de Carvalho [1,2,3]

1 ISOPlexis Centre Sustainable Agriculture and Food Technology, University of Madeira, Campus da Penteada, 9020-105 Funchal, Portugal
2 Centre for the Research and Technology of Agro-Environmental and Biological Sciences (CITAB), Inov4Agro—Institute for Innovation, Capacity Building and Sustainability of Agri-Food Production, University of Trás-os-Montes and Alto Douro, 5000-801 Vila Real, Portugal
3 Faculty of Life Sciences, University of Madeira, Campus da Penteada, 9020-105 Funchal, Portugal
* Correspondence: carla.ragonezi@staff.uma.pt; Tel.: +351-291-705-002

**Abstract:** *Beta patula* Aiton is a crop wild relative (CWR) which belongs to the Gene Pool 1b and is considered a Critically Endangered species, and is present in very specific environments, such as the Desembarcadouro islet (DI) in Ponta de São Lourenço or Chão islet (CI) in the Desertas Islands. The ISOPlexis Center (University of Madeira) has been providing continuous support for its in situ conservation by keeping a management plan of wild populations and an ex situ conservation strategy through the storage of accessions in the ISOPlexis GeneBank. The present work intends to present the spatial distribution of genetic variability and diversity in these *B. patula* populations using eight polymorphic Simple Sequence Repeat (SSR) markers. The overall results lead to the identification of three spots with a high genetic diversity. CI with a small cluster of individuals shows a genetic footprint different from DI, having unique alleles present in its population. DI has two distinct areas: the western area, with a higher individual density but with a lower genetic diversity and higher allele fixation; and the central area, with a lower individual count but with a higher genetic diversity and with the presence of unique alleles. Despite some genetic differences, the comparison of the two islets' DI and CI populations shows that they have more similarities than differences. Analysis of the Molecular Variance, based on the hierarchical cluster, showed a 9% diversity between populations, 68% among individuals, and 23% within individuals. This data will be used for the establishment of a protocol to monitor and manage *B. patula* genetic diversity under a genetic reserve, subsequently contributing to the European Genetic Reserve network implementation and the protection of this important CWR.

**Keywords:** endemism; crop wild relative; CWR; SSR; genetic reserve; in situ conservation

## 1. Introduction

Crop wild relatives (CWR) are an important category for agrodiversity that includes wild species related to crops. The Archipelago of Madeira (Portugal) holds a flora with 1204 spontaneous plant species and subspecies [1], which includes 430 CWR, and among them 264 native, 97 endemic and 69 introduced species (according to our surveys, and unpublished data), identifying this region as being a hotspot for in situ CWR conservation [2]. *Beta patula* Aiton is a CWR of beets crops (*Beta vulgaris* ssp. *vulgaris* L.) and belongs to its primary gene pool according to the Harlan and de Wet gene pool concept and its application proposed by Frese [3–5]. *B. patula* is listed in Annex II of the EU Habitats Directive [6], while the genus *Beta* is listed in Annex I of the International Treaty on Plant Genetic Resources for Food and Agriculture [7]. Beets comprise a great variety of crops meant for different purposes, such as food, fodder and agri-food industry (making up approximately 30% of the world's annual sugar production) [8,9].

The *Beta patula* species is endemic to Madeira Archipelago, occurring in the southeast region, Ponta de São Lourenço, specifically in the Desembarcadouro islet (DI) and Chão islet (CI) (the last belonging to the Desertas Islands) [10,11]. Nowadays, both islets are uninhabited and under protection as a Special Area of Conservation and a Special Protected Area under the Birds and Habitats Directives, respectively. This species has been classified by the IUCN as critically endangered (CR) [6] due to its (1) confined distribution in the archipelago of Madeira (less than 70 km$^2$), (2) its restrict occurrence within two small islets and patchy occupancy area (approximately 80,000–60,000 m$^2$ in DI and less than 12,000 m$^2$ in CI [10,12], and (3) its status as a threatened habitat due to biotic and abiotic factors [6] and the pressure of climatic changes.

Due to their great variation regarding ecological niches within apparently simple and discrete territories, islands are of particular interest for studies in ecology, biogeography, evolution [13] and the conservation of plant species [14]. Furthermore, islets are recognized as fragile ecosystems in which random factors (geography and landscape), intense environmental variation (climate) or human pressure can shape the insular flora and vegetation to a great extent, particularly because of the small plant population size [15–17].

Occurrence areas (DI and CI) are separated by a geographic barrier (ocean) and a distance of around 20 km, thus being considered two isolated populations (Figure 1). The DI population occurs in a larger occupancy area fragmented in patches of variable sizes, whereas the CI population occurs in a single, smaller occupancy area [12]. *B. patula* shares the habitat with 16 other CWR, among them two other beet CWR, *Beta vulgaris* L. ssp. *maritima* (L.) Arcang, and *Patellifolia procumbens* (C.Sm.) A.J.Scott, Ford-Lloyd and J.T.Williams [12]. This makes *B. patula* in the DI's population as the Most Appropriate Wild Population (MAWP) for in situ conservation [10,18]. Both populations are evolving in a semi-arid environment under specific drought and saline conditions [17,19]. However, these populations show dramatic differences in size, with the DI population having an estimated size of 16,906 individuals and the CI population 2917 individuals [10].

The importance of this approach arises from the need for the *B. patula* species diversity studies to support the establishment of conservation strategies contemplating in situ (genetic reserves) and ex situ (genebank) conservation [20–22]. In situ conservation, via genetic reserves, enables a more embracing genetic resource conservation, with species continuously adapting to the evolving environmental conditions and developing useful traits, avoiding allele fixation and the consequent loss of genetic diversity [23]. Useful traits can later be used in crop varieties improvement [22], using ex situ germplasm accessions periodically collected using genebank services. Modern biotechnology methods allow CWR to be a potential gene donor to crops that may result in commercially successful varieties [21]. Cultivated beets have low genetic diversity [24], and since *B. patula* belongs to the primary gene pool of *B. vulgaris*, they can become a source of gene donors [12], even though cross-pollination is randomly produced.

Nonetheless, information about in situ genetic diversity of *B. patula* is very limited [25]. Furthermore, the delimitation of a species' genetic reserve requires understanding the actual status of conservation and the dynamics of its population, the assessment of its habitat, ecogeographic and genetic baselines, and the establishment of the population's size and spatial distribution of its genetic variability [26,27]. Habitat and species ecogeographic baselines were described during the AEGRO project [25], while the populations' structure has been surveyed and monitored during the five-year execution of the Life Recover Natura Project [10]. The survey data were released through the Global Biodiversity Information Facility, GBIF [28], which allowed us to determine *B. patula*'s population size, its fluctuations, and to estimate the minimum viable size population. Previously, Nóbrega et al. [10] evaluated relevant data (biodiversity indices and demographic status) that could be used for the establishment of a protocol to monitor and manage the species populations' conservation status under a genetic reserve for *B. patula* and other CWR, contributing to the implementation of the European genetic reserve network.

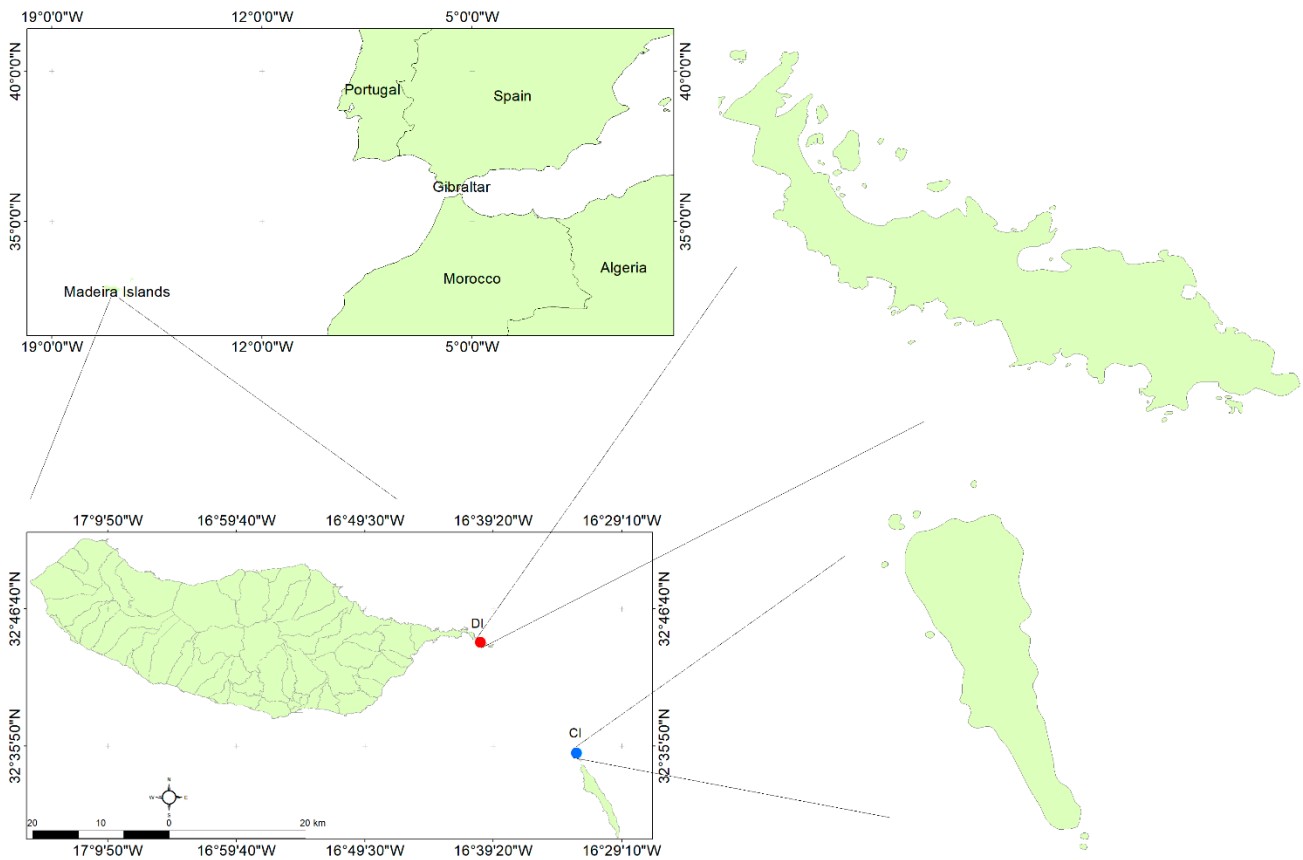

**Figure 1.** Map showing sites from Madeira where individuals of *Beta patula* were sampled, Desembarcadouro islet (DI, red dot) and Chão islet (CI, blue dot).

Although studies of genetic diversity for *B. patula* are scarce [25], there are a few studies on the phylogenetic relations between *B. patula* and other members of the *Beta* group [29,30]. Desplanque et al. [31] studied the genetic diversity of wild and cultivated *B. vulgaris* populations in France, using Restriction Fragment Length Polymorphism (RFLP) and Simple Sequence Repeat markers (SSRs). Simko et al. [32] assessed the genotypic diversity of 54 varieties of sugar beet using 30 polymorphic SSR markers, and Laurent et al. [33] identified 41 SSR markers for *Beta vulgaris* subsp. *maritima* (or wild sea beet), a species considered to be the ancestor of all cultivated beets (e.g., leaf beet, garden beet, fodder beet and sugar beet) [34], including *B. patula,* and is used in marker-assisted breeding for beets. Since the genetic relationship between DI and CI populations is not yet understood, the present work intends to assess the genetic diversity in *B. patula* populations of DI and CI and obtain the spatial distribution of genetic variability using SSR markers mostly targeting features related with useful traits for crop adaptation to abiotic conditions.

## 2. Materials and Methods

### 2.1. Study Area and Sampling Procedure

The study areas were the two occurrence sites of *Beta patula*, i.e., Desembarcadouro islet (DI) and Chão islet (CI) in the Madeira Archipelago that were previously described by Pinheiro de Carvalho et al. [11,12]. Both areas are part of the Natura 2000 network (https://ec.europa.eu/environment/nature/natura2000/, accessed on 1 June 2021). Records on the environmental conditions, the floristic composition of the areas, as well as the pattern of spatial distribution and the abundance of *B. patula*, were recorded repeatedly between 2014 and 2022 [28]. Regarding the sampling procedure, leaf samples were collected randomly from *B. patula* individuals displayed along the linear transept used to study population spatial distribution. Each sample was georeferenced (~3 m) and stored in a small paper bag

with a unique field number given. A minimum distance of 15 m between each individual sampled was considered. In total, 120 specimens, 105 from DI and 15 from CI, were sampled for DNA extraction.

### 2.2. Genetic Analysis

DNA was extracted from dried leaves of the 120 collected specimens, according to the protocol by Chao et al. [35], with modifications in step 10. Briefly, after recovering 600 μL of the supernatant, samples were washed with the addition of the same volume of chloroform: isoamyl alcohol (24:1), mixed for a few seconds, and centrifuged for 10 min at 12,000 rpm. The recovered supernatant was transferred to microtubes containing 360 μL of isopropanol, mixed thoroughly, and left to precipitate for 15 min. The first wash is used to obtain a purer sample. Modifications were also conducted in step 14, where the pellet was resuspended in 100 μL of Tris-EDTA solution, instead of ddH$_2$0, since Tris-EDTA provides better long-term conservation, and was left to dissolve overnight at 4 °C. The supernatant was recovered and stored at −20 °C. Steps 15, 16, and 17 of the procedure were skipped.

Eight polymorphic SSR markers were chosen based on their correlated traits (Table 1). DNA fragments were amplified, and PCR products were run in polyacrylamide gel electrophoresis (5% concentration). Band patterns obtained were analyzed using Fingerprinting II Informatix software (Bio-Rad Laboratories) and the resulting polymorphic band classes determined the number and size of the alleles.

**Table 1.** SSR markers designations, associated traits and bibliographic sources.

| Type | Designations | Traits | References |
|---|---|---|---|
| SSR | 2KWS (SSR2) | Leaf and root K$^+$, Leaf Ca$^{2+}$, Root Na$^+$, WSC, ECS; SC + WSC, ANC; saline and non-saline conditions. | Abbasi et al. [36] |
| SSR | BQ584037 | Phosphatidylglycerol-phosphate synthase. | McGrath et al. [37] |
| SSR | BQ588629 | BSD domain-containing protein (Pfam PF03909), root K+, WSY, WSC, SY, under saline conditions. | Abbasi et al. [36] |
| SSR | FDSB1027 | Tissue Na+ content, SY, WSY; saline responses. | Abbasi et al. [36] |
| EST-SSR | FDSB1250 | Hydrolase family protein (Pfam PF00657); GDSL esterase/lipase; WSC under non-saline conditions. | Abbasi et al. [36] |
| SSR | SB04 | Leaf Na+, Leaf K+, WSC under saline conditions. | - |
| SSR | SB13 | WSC, SC, root ANC under saline conditions. | Abbasi et al. [36] |
| SSR | SB15 | Sugar yield-related traits: SY, WSY, RY, WSC, ECS; saline responses. | Abbasi et al. [36] |

Abbreviations: EST—expressed sequence tag; WSC—white sugar content; ECS—extraction coefficient of sugar; SC—sugar content; WSY—white sugar yield; SY—sugar yield; RY—root yield; ANC—amino nitrogen content.

### 2.3. Data Treatment and Statistical Methods

Principal Coordinates Analysis (PCO [38]) was performed using the software package MVSP (Multivariate Statistical Package) version 3.1, with data processed using Gower General Similarity Coefficient and transformed using log(e). Genetic variation among and within individuals and between populations was evaluated through an AMOVA (analysis of molecular variance) using GenAlEx 6.503 [39]. The AMOVA analyses were performed using all individuals from DI and CI populations. Heatmaps were generated using ArcGIS software version 10.6.1 [40] and using IDW (Inverse Distance Weighted) method based on the observed heterozygosity. Other calculations were performed for both populations using the Population Genetics (PopGene) software version 1.31 [41]. Formulas are given as follows:

Shannon–Wiener's index (H′) [42] quantifies the uncertainty in predicting the species identity of an individual that is taken at random from the dataset. The formula is given as:

$$H' = \sum_{i=1}^{S} p_i ln \; p_i = - \sum_{i=1}^{S} L_n p_{\;i}^{\;p_i}$$

where: $S$—number of microsatellite loci assessed; $p_i$—the proportion of species $i$; $P\_i = n_i/N$; $n_i$—number of copies of a particular allele $i$; $N$—is the total number of alleles for a particular locus in the sampled population.

Observed heterozygosity (*Ho*) and the Expected heterozygosity (*He*) (Nei's algorithm [43]). Ho is the fraction of individuals in the population that are heterozygous at a given locus. The He is the fraction of heterozygotes that are expected in the population under the Hardy–Weinberg model.

$$Ho = 1 - \sum_{i=1}^{n} f \left[ A_i A_i \right]$$

where: $n$ represents the number of alleles at the locus and $i$ indicates the sequential number of that allele, $f$ indicates the frequency, and $A_i A_i$ is the homozygote frequency for each homozygote at each sequential allele.

$$He = 1 - \sum_{i=1}^{n} p_i^2$$

where: $n$ represents the number of alleles at the locus and $i$ indicates the sequential number of that allele, $p_i$ is the allele frequency for the $i$th allele, and the summation is over all available alleles.

Polymorphic information content (*PIC*) was calculated to measure the molecular markers′ ability to detect polymorphisms [44].

$$PIC = 1 - \sum_{i=1}^{n} p_i^2$$

where: $p_i^2$ is the frequency of the $i$th allele.

The fixation index ($F_{IS}$—Wright [45]), also known as the local inbreeding coefficient, defines the reduction in heterozygosity at any one level of the population hierarchy relative to any other.

$$F_{IS} = \frac{H_s - H_o}{H_s}$$

where: $H_s$, refers to the average heterozygosity of subpopulations.

## 3. Results

Regarding the SSRs results, a total number of 74 alleles were obtained, with a resulting number of 592 genotypes for the eight molecular markers. A maximum of 17 alleles were obtained for SB15, a marker linked with sugar yield-related traits and saline responses (Tables 1 and 2), and a minimum of 5 alleles for FDSB1250, with a mean number for the total dataset of 9.25 alleles (Table 2). These SSRs appear linked with the Pfam PF00657 (Tables 1 and 2), a family protein motif common to several hydrolases including GDSL serine esterases/lipases. The Polymorphism Information Content (PIC) values range from a maximum of 0.870 to a minimum of 0.406, corresponding to the 2KWS and FDSB1250 markers, respectively, with a mean value of 0.693 (Table 2).

In the Principal Coordinates Analysis (PCO), it is not possible to observe a complete separation between DI and CI individuals but three separated plant clusters in the lower quadrants (Figure 2) can be observed. Cluster I (purple circle) gathers individuals from CI (Figures 2 and 3B and S1B), Clusters II (yellow circle) gathers individuals from the western part of DI, where allele frequency is low (Figures 2, 3A and S1A), while Cluster III (green circle) gathers individuals from the center part of DI, where allele frequency is high (Figures 2, 3A and S1A). Other individuals from DI and CI form a big group distributed throughout the upper quadrants (Figure 2).

**Table 2.** SSR markers designations and associated traits.

| Molecular Marker | Linkage Group | Alleles | PIC |
|---|---|---|---|
| 2KWS (SSR2) | 2 | 11 | 0.870 |
| BQ584037 | 2 | 10 | 0.858 |
| BQ588629 | 1 | 9 | 0.662 |
| FDSB1027 | 3 | 8 | 0.670 |
| FDSB1250 | 7 | 5 | 0.406 |
| SB04 | 5 | 8 | 0.739 |
| SB13 | 3 | 6 | 0.605 |
| SB15 | 5 | 17 | 0.734 |

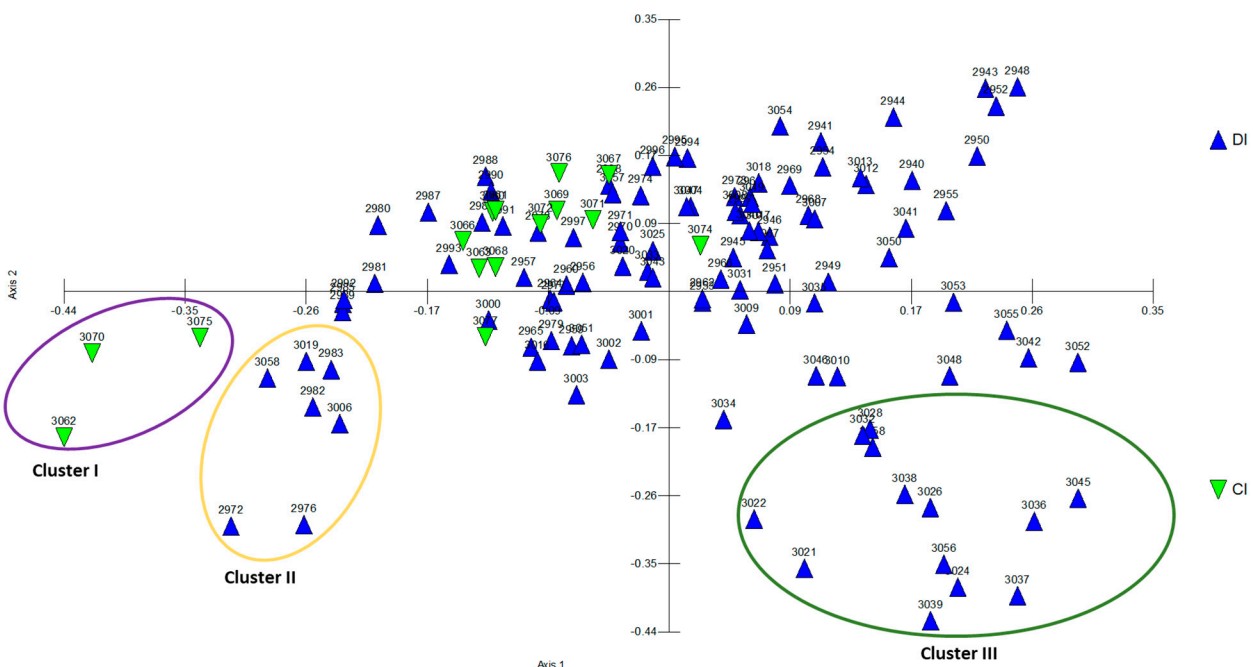

**Figure 2.** Principal Coordinates Analysis (PCO) scatterplot, showing individuals from the Chão islet and Desembarcadouro islet distributed according to genotypic data.

For the Desembarcadouro islet, Shannon–Wiener's index ($H'$) values range from 2.0976 (BQ584037) to 0.7821 (FDSB1250), and Fixation Index ($F_{IS}$) values vary from 0.9716 (FDSB1027) to 0.1575 (BQ584037) (Table 3). Both these measures ($H'$ and $F_{IS}$) are somewhat inversely relatable, since $H'$ is a diversity index, reflecting the number of different types found in our sample, and $F_{IS}$ shows the inbreeding coefficient, ranging from 0 to 1. $F_{IS}$ values of 1.0000 or close to 1.0000 mean a complete lack of heterozygotes in the population, which may be related to allele fixation at the individual level but not at the population level, which may contain different alleles for a particular locus. The expected heterozygosity for Nei's algorithm is higher for 2KWS (0.8601) and lower for FDSB1250 (0.4067) (Table 3). For the Chão islet, Shannon–Wiener's index values range from 1.7458 (BQ584037) to 0.4851 (BQ588629 and FDSB1250), and $F_{IS}$ values vary from 1.0000 (FDSB1027 and FDSB1250) to 0.1713 (BQ584037). Expected heterozygosity for Nei's algorithm is higher for BQ584037 (0.8044) and lower for BQ588629 and FDSB1250, with equal results for both markers (0.2400) (Table 3).

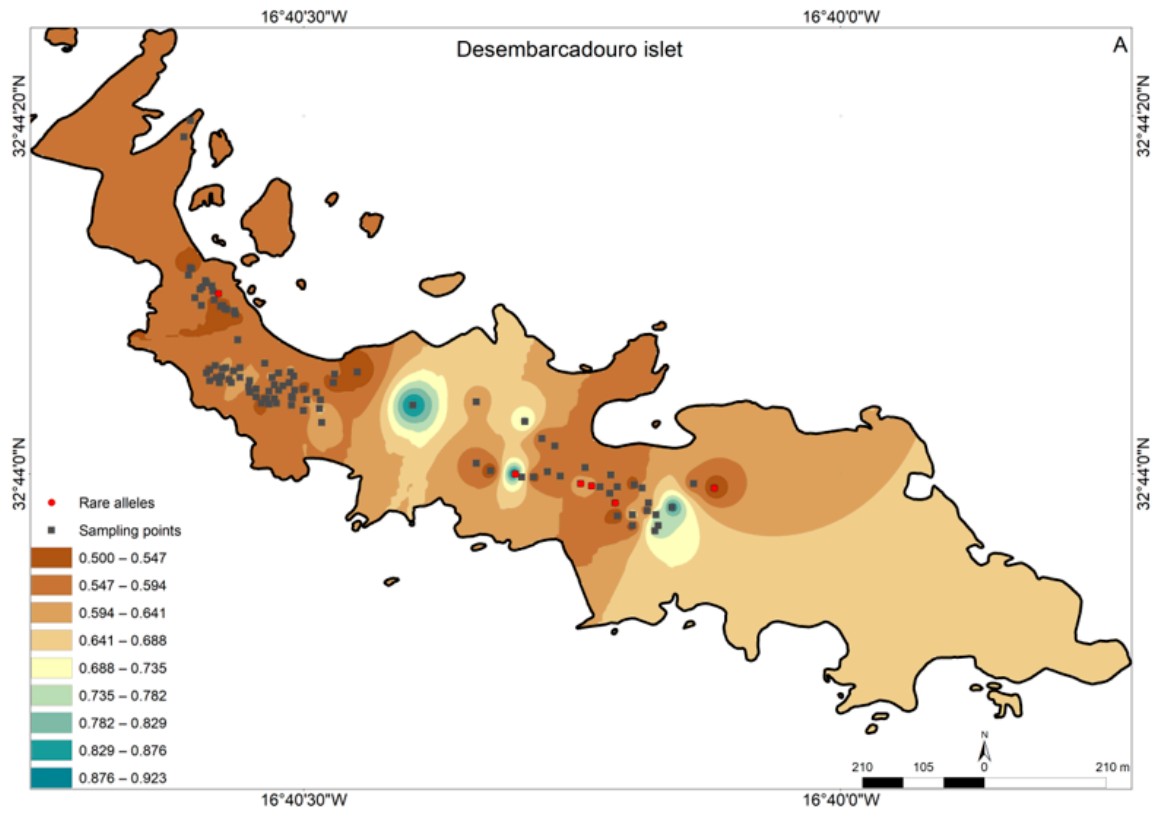

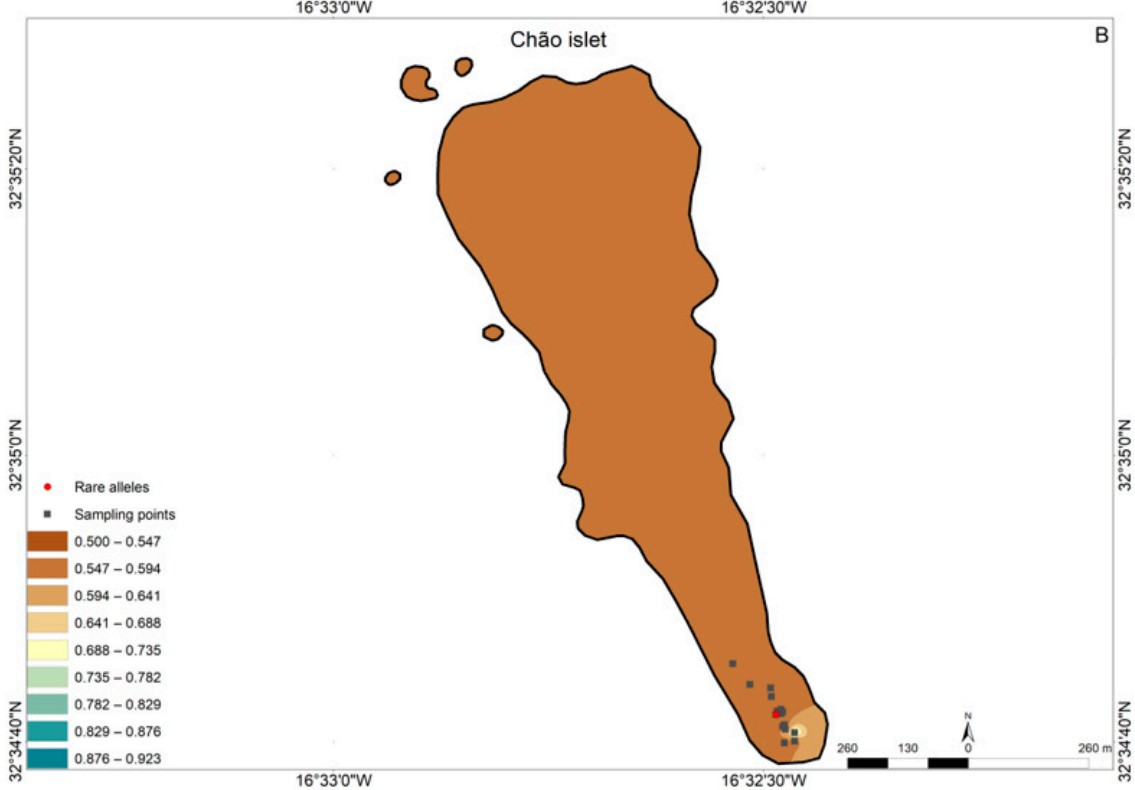

**Figure 3.** Heatmaps of Desembarcadouro islet (**A**) and Chão islet (**B**). Bluish regions in the heatmaps indicate a higher allele frequency and brownish regions indicate lower ones. Red dots indicate individuals that scored for rare alleles.

**Table 3.** Obtained values for Shannon–Wiener's index ($H'$), observed heterozygosity ($Ho$), Nei's expected heterozygosity (Nei) and fixation index ($F_{IS}$) for both studied populations.

| | Desembarcadouro Islet (DI) | | | | Chão Islet (CI) | | | |
|---|---|---|---|---|---|---|---|---|
| **Marker** | **$H'$** | **$Ho$** | **Nei** | **$F_{IS}$** | **$H'$** | **$Ho$** | **Nei** | **$F_{IS}$** |
| 2KWS | 2.0803 | 0.1048 | 0.8601 | 0.8782 | 1.6566 | 0.1333 | 0.7911 | 0.8315 |
| BQ584037 | 2.0976 | 0.7238 | 0.8591 | 0.1575 | 1.7458 | 0.6667 | 0.8044 | 0.1713 |
| BQ588629 | 1.2872 | 0.1524 | 0.6079 | 0.7493 | 0.4851 | 0.1333 | 0.2400 | 0.4444 |
| FDSB1027 | 1.3882 | 0.0190 | 0.6716 | 0.9716 | 1.0776 | 0.0000 | 0.5778 | 1.0000 |
| FDSB1250 | 0.7821 | 0.0381 | 0.4067 | 0.9063 | 0.4851 | 0.0000 | 0.2400 | 1.0000 |
| SB04 | 1.5791 | 0.0857 | 0.7264 | 0.8820 | 1.2382 | 0.1333 | 0.6867 | 0.8058 |
| SB13 | 1.0695 | 0.0857 | 0.5723 | 0.8502 | 1.4154 | 0.1333 | 0.7289 | 0.8171 |
| SB15 | 1.7787 | 0.1524 | 0.7227 | 0.7891 | 1.1711 | 0.1333 | 0.6489 | 0.7945 |
| Mean | 1.5078 | 0.1702 | 0.6784 | | 1.1593 | 0.1667 | 0.5897 | |
| St. Dev. | 0.4684 | 0.2287 | 0.1512 | | 0.4749 | 0.2108 | 0.2279 | |

Higher values for observed heterozygosity ($Ho$) are reflected in the $F_{IS}$ results, with a higher quantity of observed heterozygosity (BQ584037, for both populations) corresponding to a lower $F_{IS}$ (indicating that there is less probability for inbreeding with this trait evolved in the synthesis of regulation factors that are precursors to environmental constraints), and lower observed heterozygosity values (FDSB1027, for both populations, plus FDSB1250 for CI) corresponding to a higher $F_{IS}$ (indicating that there is a higher probability for inbreeding with these traits related to saline responses and BSD transcriptional factors). Six molecular markers stand out regarding $Ho$: four for the highest values (2KWS, BQ584037, SB15, and BQ588629), and the other two for lowest values (FDSB1027 and FDSB1250), also inversely reflected by the $F_{IS}$ values (the highest and lowest diversity resulting in the lowest and highest $F_{IS}$ values, respectively).

The Analysis of Molecular Variance (AMOVA) showed significant ($p < 0.001$) molecular variances between populations, among individuals within populations, and within individuals. The largest genetic variation (68%) was attributed to variation among individuals, while 23% of the total variation was explained by variation within individuals, and 9% was explained by variations between populations (Table 4). The $F_{ST}$ value ranges between 0 and 1, where zero indicates complete homogeneity of the allele frequencies and one describes the complete fixation of alternative alleles amongst subpopulations. Overall, the $F_{ST}$ value had a low degree of differentiation between populations (0.094), and in line with the $F_{ST}$, the $F_{IS}$ and $F_{IT}$ were high (0.749 and 0.773), meaning that the populations have a considerable degree of inbreeding.

**Table 4.** Analysis of molecular variance (AMOVA) and F-statistics between DI and CI populations.

| Source of Variation | df | SS | MS | Variance Estimated | Percentage of Variance (%) | F-Statistics | $p$ |
|---|---|---|---|---|---|---|---|
| Between populations | 1 | 19.493 | 19.493 | 0.281 | 9 | $F_{ST} = 0.094$ | <0.001 |
| Among individuals | 118 | 559.086 | 4.738 | 2.029 | 68 | $F_{IS} = 0.749$ | <0.001 |
| Within individuals | 120 | 81.500 | 0.679 | 0.679 | 23 | $F_{IT} = 0.773$ | <0.001 |
| Total | 239 | 660.079 | - | 2.990 | 100 | - | - |

df = degree of freedom, SS = sum of squares, and MS = mean squares.

The heatmaps in Figure 3 portray the genetic diversity distributed throughout *B. patula*'s geographical occurrence sites as well as the prediction of allele frequency. Bluish regions in the heatmaps indicate a higher allele frequency and brownish regions indicate lower ones. Red dots indicate specimens scored for rare alleles. In the Desembarcadouro islet (Figure 3A), a higher allele frequency was mainly observed in the islet's center, with one individual showing the highest heterozygosity (0.923), and five of the six rare alleles present being distributed through the BQ588629 (1), FDSB1027 (1), and SB15 (3) primers.

Nonetheless, a greater number of individuals were observed in the eastern region of the islet. The results range from 0.500 to 0.923 in allele frequency. Regarding the rare alleles, six were detected, with four of these present in the SB15 primer and 1 for the BQ588629 and FDSB1027 primers. All these rare markers are linked with sugar metabolism and yield, responses to saline and abiotic stress, or in the case of BQ588629, with the BSD domain-containing protein link with transcription factors, which seems to be responsive to root K+ and white sugar content, under saline conditions [36]. These alleles are mostly concentrated in the center of the ID islet (Figure 3A). The only rare allele in the western part of the islet is part of the SB15 primer, a marker also related to sugar metabolism and yield as well as responses to saline conditions, with a heterozygosity rate of 0.688. For the Chão islet, *B. patula* specimens are observed only in the southern part of the islet, and only one rare allele was detected for the 2KWS, with a heterozygosity rate of 0.813 (Figure 3B).

## 4. Discussion

Climate change is a challenge for the agri-food industry, impacting crops and diminishing production and yield. Thus, it becomes necessary to improve crop varieties, their productivity and yield features, in order to resist the main climate and abiotic constraints, e.g., the rise of temperature, drought and salinity related to aridity, which can be achieved through specific trait transference and resistance. The occurrence and evolution of *Beta patula* in a habitat, where these extreme conditions are present, make this beet CWR a valuable source of genetic material for breeding purposes.

Before the genetic analysis to evaluate the spatial distribution of genetic variability between *B. patula* populations, our concept was that CI's and DI's species occurrences represent distinct populations evolving separately. This hypothesis is supported by the fulfillment of several criteria required for the separation and evolution of species populations, such as the existence of geographic barriers and reduced probability of cross-pollination and exchange of genetic material between the two populations, due to the physical distance and ocean barrier. In detail, the reasons that could support this concept were (i) *B. patula*'s capacity for self-fertilization [46]; (ii) the 20 km distance between DI and CI occurrences, which can hamper wind pollination (the major mechanism of pollination in beets, according to Cureton et al. [47]); (iii) problematic seed dispersion by sea [47], since the *B. patula* population in CI grows at more than 50 m above the sea plateau; and finally (iv) the accidental bird seeds transportation proposed by some nature guards (personal communication), which is unlikely and needs to be confirmed by further research.

Despite some genetic differences, the comparison of the DI´s and the CI´s populations show that they have more similarities than differences. The PCO scatterplot corroborates with the Analysis of Molecular Variance (AMOVA) data, meaning that the difference between populations is not very high, and their total segregation was not possible according to the genetic data available. The PCO shows that the CI population individuals are not spatially separated from the DI's population, pointing out the absence of genetic segregation between populations. Values obtained for the different indexes (H' and Nei's) are higher for the DI's population, comparatively to the CI's, which is expected since this population has a higher size, effective population size, and a bigger occurrence area, and, consequently, a wider variation of edaphic and environmental conditions. Isolation, small size, and the lowest variation of environmental conditions can affect its fitness, creating a bottleneck in the genetic variability, with increased levels of inbreeding [48]. This inbreeding trend seems to occur in both DI and CI populations, despite the differences in population size and the greater uniformity of landscape in the CI population. Such a conclusion is supported by Table 3, where $F_{IS}$ results for CI and DI show the highest values in an equal number of molecular markers. However, DI and CI populations are different in the markers' observed heterozygosity. We hypothesize that the differences may have resulted from: (i) the founder effect of fewer specimens on the CI that could cause a higher bottleneck in population diversity; (ii) in this situation there occurred a greater alleles fixation in the genotypes, favoring those that could attribute greater adaptive capacity

concerning observed environmental conditions and abiotic stresses. Despite the small number of individuals used in the analysis, the sample is proportional to CI's estimated population sizes. The discrepancy between observed heterozygosity and Nei's expected heterozygosity values confirms the effect that small-size populations and auto-fertilization have on the reduction of available heterozygosity and, therefore, genetic diversity. *Beta patula* is an endemically critical endangered species, so its occurrence and occupancy areas and population sizes makes it very sensitive to environmental stochasticity [48], which stresses the need to know its available genetic variability in order to implement a genetic reserve management plan. This should raise awareness for the urgent and much-needed creation of protection measures for this endangered species, which now has a genetic background to sustain. These results are concordant with the results of Frese et al. [25] for genetic/geographic isolation and self-pollination.

Polymorphic information content (PIC) is a widely used parameter to evaluate the discriminatory power of molecular markers and to study the genetic diversity in populations of several taxa. PIC measures the ability of a marker to detect polymorphisms and therefore has enormous importance in the selection of molecular markers for genetic studies. Molecular markers with PIC values greater than 0.5 can be considered very informative, values between 0.25 and 0.50 are considered somewhat informative, and values lower than 0.25 are not very informative according to Botstein et al. [49]. With exception of FDSB1250, related with white sugar content under non-saline conditions [36], the molecular markers used in this show values higher than 0.5 and are therefore assumed to be good diversity indicators. PIC has been used to detect the presence of rare alleles associated with saline responses in sugar beet [36] and assess the diversity of sea populations [44]. Higher values of PIC are related to both a higher number of alleles and a more homogeneous allelic frequency distribution. This approach was used previously to assess the number of alleles, genetic pattern and differentiation of the genus *Patellifolia*, and to recommend the establishment of a genetic reserve for species conservation [50].

The Analysis of Molecular Variance (AMOVA) provides one of the most widely used frameworks for analyzing population genetic data. An $F_{ST}$ value greater than 0.15 can be considered significant in differentiating populations [51]. The values of $F_{ST} = 0.094$ and 9% of the molecular variance between populations do not support the population's genetic segregation based on available information and point out that DI and CI populations are genetically close. Similarly, to our results, Fikre et al. [52] recorded that only 10% of the variation was explained by variations among populations of varieties of *Eragrostis tef* (Zucc.) Trotter, a crop used by farmers in Ethiopia. Unlike our results, Mekbib et al. [53] observed a higher variation among populations (59.98%) and a high level of genetic differentiation ($F_{ST} = 0.596$) among three groups of aquatic macrophyte species from Asia. Some reports indicated that the level of polymorphism depends on the type of genotypes [54], the marker system used [55], the primers selected [56] and the sampling strategy [57]. Regarding island studies, large islands have a bigger area and habitat diversity, and the phylogenetic clustering shifts to phylogenetic overdispersion, while the small islands present the opposite scenario with a tendency to be more phylogenetically clustered [58].

Obtained results of the genetic diversity of DI and CI *Beta patula* populations can be applied to design an in situ conservation of this endangered species. Ottewell et al. [59] proposed a framework that identifies appropriate management strategies for threatened populations. In our case, the populations had low genetic differentiation ($F_{ST} = 0.094$), high genetic diversity ($He$) = 0.634) and high inbreeding ($F_{IS} = 0.749$). These results agree with a scenario where species populations are historically connected with previously large population-effective sizes, with a recent fragmentation or reduction in population size causing inbreeding. This species occurs in a naturally protected area of Ponta de São Lourenço and the Desertas Islands, implemented in 2015 and 1990, respectively [60]. At the same time, these occurrence places are arid or semi-arid, suffering the influence of sea salinity, high irradiation and long periods of severe drought [11]. Previously, ID and IC have been actively used by local inhabitants for the fishery, livestock and cereal production,

suffering anthropogenic pressure. Historically, ID and IC islets were known to harbor cereal crop cultivation and were exploited for agricultural purposes. Species with such a location occurring in urban and agricultural landscapes or outlier populations may be particularly susceptible to such pressure, with the decline of its population below minimal viable size.

In the Life Recover Natura Project (LIFE12 NAT/PT/000195), with the intent of recovering species and habitats within the Natura 2000 network (comprising *Ponta de São Lourenço* and *Desertas* Islands [61]), a genetic diversity study of *B. patula* was developed, aiming to identify a suitable place for a genetic reserve implementation. The genetic reserves for in situ conservation of *B. patula* should cover the maximal genetic diversity displayed in the populations and individuals in order to provide the population a viable size and to favor useful traits through the adaptation to ever-changing environmental conditions. These reserves should be set up in the protected areas and, if possible, also include other endemic and native CWR species. The genetic reserves satisfying these criteria are delineated around the MAWP, which facilitates the conservation plans and should be supported through medium- or long-term management programs, including periodical assessment of population size, genetic diversity surveys and in situ and ex situ conservation of genetic material [12,62]. Analyzing the heatmaps, regarding the genetic diversity of the geographical distribution, it is possible to ascertain the existence of a gradient distribution in DI, east to west oriented, with a wider genetic diversity concentrated in the central part of DI, where one individual showed the highest heterozygosity, and five of the six rare alleles were also found. This area is the most suitable for *B. patula* integral conservation under the establishment of a genetic reserve. However, because of the rare allele's presence, it is impossible to exclude the CI's population and the most western part of the DI from such an establishment. In these places, rare alleles for the SB15 (DI western part) and 2KWS (CI exclusive) primers were found, scoring the maximum heterozygosity rate detected. These rare alleles are associated with the FDSB1027, BQ588629, SB15 and 2KWS primers linked, namely with the tissue $Na^+$ content, white sugar yield and total sugar yield (FDSB1027), root and sugar and with sugar yield (SB15), white sugar content, leaf and root $K^+$, leaf $Ca^{2+}$, root $Na^+$, amino nitrogen content (2KWS) in the saline responses [36], and protein regulation factors under stress and non-stress conditions, pointing out the importance of genetic diversity held by the *Beta patula* populations. The overall results lead to the identification of three genetic diversity hotspots. The CI with a small cluster of individuals, shows a distinct genetic footprint different from the DI, having a unique allele present in its population. The DI has two distinct areas: the western area, with a higher individual density but with lower genetic diversity and a higher allele fixation; and the central area, with a lower individual count but with a higher genetic diversity and the presence of unique alleles. The results show that genetic information is a fundamental tool in the decision-making process that identifies appropriate management strategies for threatened populations, for example, the establishment of genetic reserves, where an array of factors must be considered. Because the occurrence of *Beta patula* is so small in area and number of individuals, but with such distinct genetic traits, a special genetic reserve protection status for both islets should be considered. Additionally, genetic management strategies for populations with a high diversity and gene flow, but with the risk of population declination due to inbreeding depression in the long term, should be managed to reduce breeding between genetically related individuals. Such management actions should include, for example, the facilitation of pollen/seem immigration, active introduction of genetic diversity as well as the management of pollinator/seed disperser populations where these have been perturbed [59].

## 5. Final Remarks

Crop wild relatives' specific genes carry information resulting from evolutionary forces mixed with external influences, since many of them have evolved in harsher environments [29]. Such conclusions, once again, underpin the results and conclusions obtained

by Frese et al. [25] about the genetic reserve, including both *B. patula* populations. Insular ecosystems are particularly at risk with upcoming changes in natural resource availability, and in Madeira's case, allied to geographical restrictions, the scenario may demonstrate higher vulnerability [63]. The determination of the population's variability is a necessary task, leading to the selection of the sites for integral conservation under genetic reserves, complementing previous works of distribution data resulting from the AEGRO project [64]. This is the start of *B. patula*'s genetic information monitorization, safeguarding for climate changes that may reduce the species' distribution and the possibility of survival. The species' genetic singularity and its importance for food security reinforces our efforts to support protection status through the implantation of a genetic reserve and corresponding management plan [25] aligned with the European strategy for in situ conservation of CWR [22].

**Supplementary Materials:** The following supporting information can be downloaded at: https://www.mdpi.com/article/10.3390/agriculture13010027/s1, Figure S1. Distribution maps of individuals from Desembarcadouro islet (A) and Chão islet (B).

**Author Contributions:** Conceptualization: M.I.L., H.N., M.Â.A.P.d.C.; Data curation: C.R., H.N., M.Â.A.P.d.C.; Investigation: C.R., H.N., M.I.L., J.G.R.d.F.; Methodology: C.R., H.N., M.I.L., J.G.R.d.F.; Writing—original draft preparation: M.I.L., H.N.; Writing—review and editing; C.R., H.N., F.L.M., M.Â.A.P.d.C.; Supervision: M.Â.A.P.d.C.; Funding acquisition: H.N., J.G.R.d.F., M.Â.A.P.d.C. All authors have read and agreed to the published version of the manuscript.

**Funding:** This research was funded by *Programa Operacional* Madeira 14–20, Portugal 2020, and the European Union through the European Regional Development Fund, grant number M1420-01-0145-FEDER-000011 [CASBio]; the Project LIFE RECOVER Natura—LIFE12 NAT/PT/000195; and by the *Agência Regional para o Desenvolvimento da Investigação, Tecnologia e Inovação*, Portugal 2020 and the European Union through the European Social Fund [grant number M1420-09-5369-FSE000002, ARDITI].

**Institutional Review Board Statement:** Not applicable.

**Data Availability Statement:** Not applicable.

**Acknowledgments:** The authors acknowledge the support by National Funds FCT-Portuguese Foundation for Science and Technology under the projects UIDB/04033/2020 and UIDP/04033/2020.

**Conflicts of Interest:** The authors declare no conflict of interest.

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
