# Peer review of "Distribution of Genetic Diversity in Beta patula Aiton Populations from Madeira Archipelago, Portugal"

_agriculture, doi:10.3390/agriculture13010027_

Round 1
Reviewer 1 Report
The authors have investigated distribution of genetic diversity in Beta patula Aiton populations from Madeira Archipelago in Portugal. I find that the conceptualisation of the research is good, and that influence of that research is huge in agriculture and conservation biology. However, the manuscript in the present form is not appropriate for publication, and I suggest major revision. All comments are provided in attached the pdf document of the manuscript. I look forward to seeing revised version of the manuscript.

Author Response
Response to Reviewer 1 Comments
Manuscript ID: agriculture-2052194
Type of manuscript: original article
Title: Distribution of genetic diversity in Beta patula Aiton populations from Madeira Archipelago, Portugal
Authors: Carla Ragonezi, Humberto Nóbrega, Maria Inês Leite, José G. R. de Freitas, Fabrício Lopes Macedo, Miguel Â. A. Pinheiro de Carvalho
Point 1: The authors have investigated distribution of genetic diversity in Beta patula Aiton populations from Madeira Archipelago in Portugal. I find that the conceptualisation of the research is good, and that influence of that research is huge in agriculture and conservation biology. However, the manuscript in the present form is not appropriate for publication, and I suggest major revision. All comments are provided in attached the pdf document of the manuscript. I look forward to seeing revised version of the manuscript.
Response 1: The authors appreciate the reviewer’s comments about the manuscript. The comments that concern corrections/changes/observations can be visualized in the resubmitted version. The comments that were questions or observations that needed response were addressed properly below.
Response 2: The word “biodiversity” was replaced. The replacement was done to better describe the context. The idea is to introduce what makes de plants differentiate themselves in the niches and how this experience can be used in different areas of study.
Point 3: Line 149, Why did you have that ratio?
Response 3: The specimens’ ratio is related to the number of individuals present in the sampling area at the collection time. For CI, the population size was constricted to a small area with a few individuals, which is limited.
Point 4: Line 152, You should list step 11, 12, and 13.
Response 4: These steps were not listed because they did not have any modifications.
Point 5: Line 321, This is true for H' and Nei's index but not for observed heterozigosity (DI 0.1048, CI 0.1333; please see the Table 3).
Response 5: The sentence was corrected by removing the Ho value.
Point 6: Line 329, I do not agree with that. According reference 7, estimated population size in DI was 16 906, and in CI was 2 917. In the present study is analyzed 120 specimens (105 DI and 15 CI).
Response 6: The numbers from reference 7, are estimated values collected during a 4-year period. Mainly IC suffers great fluctuations in its estimated population from one year to another. Thus, could be expected that the estimated proportions are not the same as the ones from this study where individuals were collected during a time point.
Point 7: Line 360, In your study, the largest genetic variation was among individuals (68%).
Response 7: The authors are not sure about this comment. The 60% inside the parenthesis is from the cited article, and not from our results.
Point 8: You should check again the reference in the reference list. I found some mistakes and i would like that you check all references again and correct them.
Response 8: The authors checked all the references. Tracking, in that case, is not visible since we used Mendeley.

Reviewer 2 Report
This study presents the genetic diversity in a crop wild relative plant, which is potentially interesting to the readers.
However, the presentation of the methods and results need to be improved.
-Could the authors discuss more about the population difference between DI and CI? In Figure3B, only small part of the islet is sampled. Is the map correctly indicated the allele frequency where nothing is sampled?
In addition, the figure legend “allele frequency” in Figure 3 looks not clearly presented. The blue squares seem like sampling points.
-The calculation methods should be clearly cited, such as that of the Polymorphic information content (PIC).
-The terms Ho, He, Nei, Ob. He should be consistent with each other in page 5, page 7, and throughout the manuscript.
-There are inconsistent descriptions and formats in the manuscript. For example, the italic of scientific names and nouns is inconsistent. The blank key spaces need to be checked.
-page 8, line 265, ID islet looks incorrect. Page 11, line 343, ID and IC islets?
Author Response
Response to Reviewer 2 Comments
Manuscript ID: agriculture-2052194
Type of manuscript: original article
Title: Distribution of genetic diversity in Beta patula Aiton populations from Madeira Archipelago, Portugal
Authors: Carla Ragonezi, Humberto Nóbrega, Maria Inês Leite, José G. R. de Freitas, Fabrício Lopes Macedo, Miguel Â. A. Pinheiro de Carvalho
The authors appreciate the reviewer’s comments about the manuscript. The comments were addressed properly below and can be visualized in the resubmitted version.
Point 1: This study presents the genetic diversity in a crop wild relative plant, which is potentially interesting to the readers. However, the presentation of the methods and results need to be improved.
Response 1: The authors appreciate the reviewer´s comments and overall observations.
Point 2: Could the authors discuss more about the population difference between DI and CI? In Figure3B, only small part of the islet is sampled. Is the map correctly indicated the allele frequency where nothing is sampled? In addition, the figure legend “allele frequency” in Figure 3 looks not clearly presented. The blue squares seem like sampling points.
Response 2: The population difference between DI and CI was further discussed in the discussion section. The authors appreciate the reviewer´s comments and replaced the figure legend from “allele frequency” with “sampling points”. Regarding Figure 3B, the colors on the heatmap are predictions based on data collected on the field made by the ArcGis software. The heatmaps help in the prediction of high genetic diversity areas, thus, helps in the implementation of possible reserve management actions.
Point 3: The calculation methods should be clearly cited, such as that of the Polymorphic information content (PIC).
Response 3: All calculation methods are presented in the methodology section, and PIC, for example, can be found in line 201.
Point 4: The terms Ho, He, Nei, Ob. He should be consistent with each other in page 5, page 7, and throughout the manuscript.
Response 4: The authors proceeded to consistency throughout the manuscript.
Point 5: There are inconsistent descriptions and formats in the manuscript. For example, the italic of scientific names and nouns is inconsistent. The blank key spaces need to be checked.
Response 5: The authors proceeded to consistency throughout the manuscript.
Point 6: page 8, line 265, ID islet looks incorrect. Page 11, line 343, ID and IC islets?
Response 6: The denomination is correct, is ID and IC islets.

Round 2
Reviewer 1 Report
Dear author,
The revised version of the manuscript is significantly improved, but I found that minor revision is required. All comments are attached in the pdf document of your manuscript. Also, in supplementary file you should write the name of species in italic, but names of islets should not write in italic.

Author Response
Response to Reviewer 1 Comments
Manuscript ID: agriculture-2052194
Type of manuscript: original article
Title: Distribution of genetic diversity in Beta patula Aiton populations from Madeira Archipelago, Portugal
Authors: Carla Ragonezi, Humberto Nóbrega, Maria Inês Leite, José G. R. de Freitas, Fabrício Lopes Macedo, Miguel Â. A. Pinheiro de Carvalho
Point 1: The revised version of the manuscript is significantly improved, but I found that minor revision is required. All comments are attached in the pdf document of your manuscript. Also, in supplementary file you should write the name of species in italic, but names of islets should not write in italic.
Response 1: The authors appreciate the reviewer’s comments about the manuscript. The comments that concern corrections/changes/observations can be visualized in the resubmitted version. The comments that were questions or observations that needed response were addressed properly below.
Point 2: Reference: In some places the year of publication is written in bold, but in some places is not. Please, uniform that.
Response 2: The references were done according to the instructions for the authors of the Agriculture Journal. In some cases, the date should not be bold.
